# Bioactive exometabolites drive maintenance competition in simple bacterial communities

John L. Chodkowski,[1] Ashley Shade[2]

**ABSTRACT** During prolonged resource limitation, bacterial cells can persist in metabolically active states of non-growth. These maintenance periods, such as those experienced in stationary phase, can include upregulation of secondary metabolism and release of exometabolites into the local environment. As resource limitation is common in many environmental microbial habitats, we hypothesized that neighboring bacterial populations employ exometabolites to compete or cooperate during maintenance and that these exometabolite-facilitated interactions can drive community outcomes. Here, we evaluated the consequences of exometabolite interactions over the stationary phase among three environmental strains: *Burkholderia thailandensis* E264, *Chromobacterium subtsugae ATCC 31532*, and *Pseudomonas syringae* pv. *tomato* DC3000. We assembled them into synthetic communities that only permitted chemical interactions. We compared the responses (transcripts) and outputs (exometabolites) of each member with and without neighbors. We found that transcriptional dynamics were changed with different neighbors and that some of these changes were coordinated between members. The dominant competitor *B. thailandensis* consistently upregulated biosynthetic gene clusters to produce bioactive exometabolites for both exploitative and interference competition. These results demonstrate that competition strategies during maintenance can contribute to community-level outcomes. It also suggests that the traditional concept of defining competitiveness by growth outcomes may be narrow and that maintenance competition could be an additional or alternative measure.

**IMPORTANCE** Free-living microbial populations often persist and engage in environments that offer few or inconsistently available resources. Thus, it is important to investigate microbial interactions in this common and ecologically relevant condition of non-growth. This work investigates the consequences of resource limitation for community metabolic output and for population interactions in simple synthetic bacterial communities. Despite non-growth, we observed active, exometabolite-mediated competition among the bacterial populations. Many of these interactions and produced exometabolites were dependent on the community composition but we also observed that one dominant competitor consistently produced interfering exometabolites regardless. These results are important for predicting and understanding microbial interactions in resource-limited environments.

**KEYWORDS** antibiotic production, biosynthetic gene clusters, *Burkholderia*, *Chromobacterium*, *Pseudomonas*, DC3000, synthetic community, SynCom, metabolomics, RNAseq

Address correspondence to Ashley Shade, ashley.shade@cnrs.fr.

The authors declare no conflict of interest.

See the funding table on p. 24.

Bacteria interact with other bacteria and their environment within complex, multi-species communities. Bacterial interactions rely on the ability to sense and respond to both biotic and abiotic stimuli (1, 2). These stimuli include physical, chemical, or molecular cues, and can alter bacterial behaviors (3, 4), and, ultimately, can also

alter community functioning (5, 6). It is expected that interspecies interactions play an important role in shaping microbial community dynamics (7). However, multiple stimuli in the environment make it difficult to disentangle the separate influences of abiotic versus biotic stimuli on microbial community dynamics (8). Therefore, efforts to characterize and distinguish community responses to biotic stimuli, such as those that facilitate interspecies interactions, will provide insights into the specific roles that microbial interactions play in shaping their communities (9).

Interspecies interactions can be facilitated through small molecules (10). Extracellular small molecules are collectively referred to as exometabolites (11–13). Depending on the exometabolite produced, these molecules can mediate interspecies interactions that range from competitive to cooperative (14). Of these interaction types, competition has been shown to have a major influence in structuring microbial communities (15–17). Thus, competitive interactions that are mediated by exometabolites are also expected to influence microbial community dynamics. In addition, different types of exometabolites can be employed by bacteria to gain the advantage in both exploitative (e.g., nutrient scavenging) and interference (direct cell damage) categories of competition.

Traditionally, competition has been viewed through the lens of resource acquisition (18). In previous studies, competitiveness is modeled with respect to yield given resource consumption and growth (19, 20). However, competition for *survival* or *maintenance* may be just as important as competition for yield, especially during periods of resource limitation (21, 22). Competition during maintenance is likely common in environments that experience relatively long periods of nutrient famine punctuated by short periods of nutrient influx, for example, as in soils, sequencing batch reactors, and the gut (23–26). The stationary phase of a bacterial growth curve falls within this context of growth cessation, and pulses of nutrients may be transiently available as cells die and lyse (necromass), while the total population size remains stagnant. The stationary phase is often coordinated with a metabolic shift to secondary metabolism (27, 28). Therefore, an effective "maintenance" competitor may produce bioactive exometabolites, like antibiotics, which are often produced because of secondary metabolism. Bacteria can activate biosynthetic gene clusters (BSGCs) to produce bioactive exometabolites (29). The activation of BSGCs is closely tied to stress responses, suggesting that bacteria can sense the stress of competition (30, 31). While it is known that certain exometabolites can trigger BSGC upregulation and, more generally, alter transcription (32), there is much to understand about the outcomes of interspecies interactions for BSGCs in multi-member microbial communities.

Here, we build on our previous research to understand how exometabolite-mediated interactions among bacterial neighbors contribute to community outcomes in a simple, three-member community (Table 1). These three members are commonly associated with terrestrial environments (soils or plants) and were chosen because of reported (33) and observed interspecies exometabolite interactions in the laboratory. We used a synthetic community (SynCom) approach (34) by applying our previously described transwell system (35), which allowed for the evaluation of "community goods" within a media reservoir that was shared among members. The members' populations were physically isolated by membrane filters at the bottom of each transwell but could

**TABLE 1** Bacterial members used in the synthetic community (SynCom) system

| Member | Burkholderia thailandensis E264 | *Chromobacterium subtsugae* ATCC 31,532[a] | Pseudomonas syringae pv. tomato DC3000 |
|---|---|---|---|
| *Genome size (Mb)* | 6.72 | 4.76 | 6.54 |
| *Family* | *Burkholderiaceae* | *Neisseriaceae* | *Pseudomonadaceae* |
| No. of CDSs[b] | 5,639 | 4,393 | 5,576 |
| *Chromosomes* | 2 | 1 | 1 |
| *Plasmids* | 0 | 0 | 2 |
| *ATCC* | 700388 | 31532 | BAA-871 |
| *Reference* | 37 | 38 | 39 |

[a]This member in previous work (35, 36) was referred to as *Chromobacterium violaceum*. This member has been reclassified (37).
[b]CDSs, coding sequences.

interact chemically *via* the reservoir. In our prior work, we investigated each member's exometabolites and transcription over a stationary phase, and the objective was to understand monoculture responses (in minimal glucose media) before assembling the more complex two- and three-member communities. We previously found that each member in monoculture produced a variety of exometabolites in the stationary phase, including bioactive molecules involved in competition (36). In this work, we build two- and three-member communities to ask: How do members interact *via* exometabolites in simple communities during maintenance (stationary phase), and what are the competitive strategies and outcomes of those interactions? What genetic pathways, molecules, and members drive the responses?

We found that *B. thailandensis* had a major influence on the transcriptional responses of both *C. subtsugae* and *P. syringae* and that this influence could be attributed to an increase in both interference and exploitative competition strategies. These findings show that diverse competitive strategies can be deployed even when bacterial neighbors are surviving rather than exponentially growing. Therefore, we suggest that contact-independent, exometabolite-mediated interference and exploitation are important competitive strategies in resource-limited environments and support the non-yield outcome of maintenance.

## MATERIALS AND METHODS

### Bacterial strains and culture conditions

We selected three environmental bacterial strains for the SynCom experiments that were originally isolated from various plant/soil habitats and that had prior evidence of exometabolite interactions among them in the laboratory (Table 1; 33, 37–40). Freezer stocks of *B. thailandensis*, *C. subtsugae*, and *P. syringae* were plated on half-concentration Trypticase soy agar (TSA50) at 27°C for at least 24 h. Members were inoculated in 7 mL of M9—0.2% glucose medium and grown for 16 h at 27°C, 200 rpm. Cultures were then diluted into 50 mL M9—0.2% glucose medium such that the exponential growth phase was achieved after 10 h of incubation at 27°C, 200 rpm. Members were diluted in 50 mL M9 glucose medium to target ODs (*B. thailandensis* 0.3 OD, *C. subtsugae*: 0.035 OD, *P. syringae* 0.035 OD). The high initial OD for *B. thailandensis* was necessary such that the stationary phase would be achieved by all members within a 2-h window after 24-h incubation in the transwell plate. The glucose concentration in the final dilution varied upon community membership—0.067% for monocultures, 0.13% for pairwise cocultures, and 0.2% for the three-member community. For each member, 48 mL of diluted culture was transferred as 4 mL aliquots in 12, 5 mL Falcon tubes to more efficiently prepare replicate transwell plates.

### Synthetic community experiments

Transwell plate preparation was performed as previously described (35). Briefly, we used sterile filter plates with 0.22-µm-pore polyvinylidene difluoride (PVDF) filter bottoms (Millipore MAGVS2210). Prior to use, filter plates were washed three times with sterile water using a vacuum apparatus (NucleoVac 96 vacuum manifold; Clontech Laboratories). The filter of well H12 was removed with a sterile pipette tip and tweezer, and 31 mL of M9 glucose medium was added to the reservoir through well H12. The glucose concentration in the reservoir varied upon community membership—0.067% for monocultures, 0.13% for pairwise cocultures, and 0.2% for the three-member community. Glucose concentration was adjusted to plate occupancy (e.g., three-member communities had a higher number of wells occupied than two- or one-member). Our aim was for each member to achieve the stationary phase at similar times across all conditions to compare transcripts and exometabolites under similar growth trajectories. In other words, available resources were standardized while keeping the well occupancy for each member constant. With this design, transcripts and exometabolites in cocultures

that deviated from those in monocultures could be attributed to interspecies interactions and not complicated by offset in member growth trajectories across the experimental conditions.

Each well was filled with 130 µL of culture or medium (prepared as described, above; see Materials and Methods section: Bacterial strains and culture conditions). For each plate, a custom R script [RandomArray.R (see the script at https://github.com/ShadeLab/PAPER_Chodkowski_mSystems_2017/blob/master/R_analysis/RandomArray.R)] was used to randomize community member placement in the wells so that each member occupied a total of 31 wells per plate. In total, there were seven community conditions—three monocultures, three pairwise cocultures, and the three-member community. Each member occupied 31 wells per plate regardless of experimental condition. Thus, "baseline" exometabolites could be determined in the monocultures, and then deviations in exometabolite abundance or detection in the cocultures could be attributed to interspecies interactions. A time course was performed for each replicate. The time course included an exponential phase time point (12.5 h) and five time points assessed every 5 h over a stationary phase (25–45 h). Four biological replicates were performed for each community condition for a total of 28 experiments. For each experiment, six replicate filter plates were prepared for destructive sampling for a total of 168 transwell plates.

Filter plates were incubated at 27°C with gentle shaking (~0.32 rcf). For each plate, a custom R script [RandomArray.R (see the script at https://github.com/ShadeLab/PAPER_Chodkowski_mSystems_2017/blob/master/R_analysis/RandomArray.R)] was used to randomize wells for each organism assigned to RNA extraction (16 wells) and flow cytometry (five wells). The following procedure was performed for each organism when a transwell plate was destructively sampled: (i) wells containing spent culture assigned to RNA extraction were pooled (~100 µL/well) into a 1.7 mL microcentrifuge tube and flash frozen in liquid nitrogen and stored at −80 until further processing. (ii) 20 µL from wells assigned for flow cytometry were diluted into 180 µL Tris-buffered saline [TBS; 20 mM Tris, 0.8% NaCl (pH 7.4)]. In community memberships where *P. syringae* was arrayed with *B. thailandensis*, *P. syringae* had a final dilution of 70-fold in TBS. In community memberships where *P. syringae* was arrayed in monoculture or coculture with *C. subtsugae*, *P. syringae* had a final dilution of 900-fold in TBS. Final dilutions for *B. thailandensis* and *C. subtsugae* were one 300-fold and one 540-fold, respectively. Each member was diluted differently to achieve a suitable events/second range on the flow cytometer for accurate cell counting. Populations were then stained and analyzed on the flow cytometer for live/dead counting (see Supplementary Methods). (iii) Spent medium (~31 mL) from the shared reservoir was transferred to 50 mL conical tubes, flash-frozen in liquid nitrogen, and stored at −80°C prior to metabolite extraction.

## Transcriptomics

### Quality filtering and differential gene expression analysis

RNA extraction, sequencing, quality control, and count matrix generation were performed as previously published [(36), see Supplementary Methods]. Count matrices for each member were quality filtered in two steps: genes containing 0 counts in all samples were removed, and genes with a transcript count of ≤10 in more than 90% of samples were removed. DESeq2 (41) was used to extract size factor and dispersion estimates. These estimates were used as external input into ImpulseDE2 for the analysis of differentially regulated genes (42). ImpulseDE2 determines differential expression by comparing longitudinal count data sets. Case-control (Cocultures-monoculture control) analyses were analyzed to identify genes with differences in temporal regulation at an FDR-corrected threshold of 0.01. Genes that passed the FDR threshold were further filtered for genes that had at least one time point with a log2 fold-change (LFC) >= 1 or <= −1. Thus, we defined differentially expressed genes (DEGs) as genes that met both the FDR-corrected and LFC thresholds. For each member, differences in gene regulation

between the three coculture conditions were visualized with Venn diagrams using the VennDiagram package (43).

Differentially expressed genes were first determined by comparing each coculture condition to the monoculture control and applying an LFC threshold (see above). We then determined a second set of DEGs by comparing pairwise cocultures to each other. ImpulseDE2 case-control analyses were performed as follows: *B. thailandensis* coculture with *C. subtsugae* (case) compared to *B. thailandensis* coculture with *P. syringae* (control), *C. subtsugae* coculture with *B. thailandensis* (case) compared to *C. subtsugae* coculture with *P. syringae* (control), and *P. syringae* coculture with *B. thailandensis* (case) compared to *P. syringae* coculture with *C, subtsugae* (control). Genes that passed the FDR-corrected threshold of 0.01 based on ImpulseDE2 analysis and had at least one time point with an LFC of >= 1 or <= −1 represented coculture-specific DEGs. The DEGs determined from monoculture comparisons and coculture comparisons were then categorically grouped using Clusters of Orthologous Groups (COG).

## COG analysis

Protein fasta files were downloaded from NCBI and uploaded to eggNOG-mapper v2 (http://eggnog-mapper.embl.de/) to obtain COGs. The DEGs determined from ImpulseDE2 and LFC thresholds were categorized as upregulated or downregulated based on temporal expression patterns. DEGs with consistent positive LFC throughout all stationary phase time points were categorized as upregulated. DEGs with consistent negative LFC throughout all stationary phase time points were categorized as downregulated. These DEGs were then assigned to COGs, grouped based on temporal up/downregulation patterns, and plotted using ggplot2 (44).

## Principal coordinates analysis and statistics

Normalized gene matrices were extracted from DESeq2 and filtered to only contain DEGs (coculture to monoculture comparisons) based on our previously described definition. A variance-stabilizing transformation was performed on normalized gene matrices using the rlog function in DESeq2. A distance matrix based on the Bray-Curtis dissimilarity metric was then calculated on the variance-stabilized gene matrices and principal coordinates analysis was performed using the R package vegan (45). Principal coordinates were plotted using ggplot2. Coordinates of the first two PCoA axes were used to perform PROTEST analysis using the PROTEST function in vegan. Dissimilarity matrices were used to perform PERMANOVA and variation partitioning using the adonis and varpart functions in vegan, respectively. The RVAideMemoire package (46) was used to perform post hoc pairwise PERMANOVAs. Lastly, distances were extracted from the Bray-Curtis dissimilarity matrix that compared each coculture condition to the monoculture condition at each time point within each member. These distances were used to produce time series distance plots.

## BSGC analysis

NCBI accession numbers were uploaded to the antiSMASH 6 beta bacterial version (47) to identify genes involved in BSGCs using default parameters. Where possible, literature-based evidence and BSGCs uploaded to MIBiG (48) were used to better inform antiSMASH predictions. Log2 fold-changes (LFCs) were calculated for all predicted biosynthetic genes within each predicted cluster by comparing coculture expression to monoculture expression at each time point. Average LFCs were calculated from all predicted biosynthetic genes within a predicted BSGC at each time point. Temporal LFC trends were plotted using ggplot2. An upregulated BSGC was defined as a BSGC that had at least two consecutive time points in the stationary phase with an LFC > 1.

## Network analysis

Unweighted co-expression networks were created from quality-filtered and normalized expression data. Networks were generated for pairwise cocultures containing *B. thailandensis*. First, data were quality filtered as previously described (see methods section: *Quality filtering and differential gene expression analysis*). Then, normalized expression data were extracted from DESeq2. Twenty-three and twenty-four RNA-seq samples from each member were used for network analysis in the *B. thailandensis-C. subtsugae and B. thailandensis-P. syringae* cocultures, respectively (23/24 samples/member; six time points, four biological replicates). Only 23 samples were used in the *B. thailandensis-C. subtsugae* network analysis because RNA-seq failed for *C. subtsugae* at 45 h, biological replicate 2. Interspecies networks were then inferred from the expression data using the context likelihood of relatedness (49) algorithm within the R package Minet (50). Gene matrices for each coculture pair were concatenated to perform the following analysis. Briefly, the mutual information coefficient was determined for each gene pair. To ensure robust detection of co-expressed genes, a resampling approach was used as previously described (51). Then, a Z-score was computed on the mutual information matrix. A Z-score threshold of 4.5 was used to determine an edge in the interspecies network. Interspecies networks were uploaded into Cytoscape version 3.7.1. for visualization, topological analysis, and enrichment analysis (52).

Gene annotation and gene ontology (GO) files were obtained for *B. thailandensis*, *P. syringae,* and *C. subtsugae* for enrichment analyses. For *B. thailandensis,* annotation and ontology files were downloaded from the Burkholderia Genome Database (https://www.burkholderia.com). For *P. syringae*, annotation and ontology files were downloaded from the Pseudomonas Genome Database (http://www.pseudomonas.com/strain/download). Annotation and ontology files for *C. subtsugae* were generated using Blast2GO version 5.2.5 (53). InterProScan (54) with default parameters was used to complement gene annotations from *C. subtsugae*. GO terms were assigned using Blast2GO with default parameters. In addition, genes involved in secondary metabolism were manually curated and added to these files as individual GO terms. These genes were also used to update the GO term GO:0017000 (antibiotic biosynthetic process), composed of a collection of all the biosynthetic genes. (see Materials and Methods section: *Biosynthetic gene cluster (BSGC) analysis*).

Topological analysis was performed as follows: Nodes were filtered from each coculture network to only select genes from one member at a time. The GLay community cluster function in Cytoscape was used to determine intra-member modules. Functional enrichment analysis was then performed on the modules using the BiNGO package (55) in Cytoscape.

To determine interspecies co-regulation patterns, we filtered network nodes that contained an interspecies edge. Functional enrichment analysis was performed on the collection of genes containing interspecies edges for each member using the BiNGO package in Cytoscape. Then, we selected all genes contained within modules of interest (e.g., *B. thailandensis* modules containing either thailandamide or malleilactone genes in the *B. thailandensis-C. subtsugae* coculture network or *B. thailandensis-P. syringae* coculture network, respectively) in Cytoscape. Node selection was extended by selecting the first neighbors of the selected nodes. This resulted in interspecies edges. The resultant nodes were transformed into a circular layout and exported for manual edits in InkScape. The biosynthetic gene cluster organization of thailandamide and malleilactone was obtained from MIBig and drawn in InkScape.

Protein sequences from an interspecies gene of interest (CLV_2968) within a network module that also contained thailandamide genes from the *B. thailandensis-C. subtsugae* network and an interspecies gene of interest (PSPTO_1206) within a network module that also contained malleilactone genes from the *B. thailandensis-P. syringae* network was obtained. A protein blast for each protein was run against *B. thailandensis* protein sequences. *B. thailandensis* locus tags were extracted from the top blast hit from each

run. Normalized transcript counts for these four genes of interest were plotted in R. Time course gene trajectories were determined using a loess smoothing function.

## Metabolomics

### *LCMS, feature detection, and quality control*

Standard operating protocols were performed at the Department of Energy Joint Genome Institute as previously described (36). MZmine 2 (56) was used for feature detection and peak area integration as previously described (36). Select exometabolites were identified in MZmine 2 by manual observation of both MS and MS/MS data. We extracted quantities of these identified exometabolites for ANOVA and Tukey HSD post-hoc analysis in R. We filtered features in three steps to identify coculture-accumulated exometabolites. The feature-filtering steps were performed as follows on a per-member basis: (i) retain features where the maximum peak area abundance occurred in any of the coculture communities ; (ii) a noise filter, the minimum peak area of a feature from a replicate at any time point needed to be three times the maximum peak area of the same feature in one of the external control replicates, was applied; and (iii) coefficient of variation (CV) values for each feature calculated between replicates at each time point needed to be less than 20% across the time series.

Four final feature data sets from polar and nonpolar analyses in both ionization modes were analyzed in MetaboAnalyst 5.0 (57), as reported in our prior work (36, see Supplementary Methods). In addition, exometabolites categorized as primary metabolites were identified according to Metabolomics Standards Initiative (MSI) level 1 criteria (58), as reported in our prior work (36, see Materials and Methods).

### *Principal coordinates analysis and statistics*

A distance matrix based on the Bray-Curtis dissimilarity metric was used to calculate dissimilarities between exometabolite profiles. Principal coordinates analysis was performed using the R package vegan. Principal coordinates were plotted using ggplot2. Coordinates of the first two PCoA axes were used to perform Protest analysis using the protest function in vegan. Dissimilarity matrices were used to perform PERMANOVA and variation partitioning using the adonis and varpart functions in vegan, respectively. The RVAideMemoire package was used to perform post hoc pairwise PERMANOVAs. Monoculture controls were removed to focus on coculture trends.

## RESULTS

### Overview

Our major data types included both transcriptomics and metabolomics, and we integrate these to interpret SynCom dynamics and interactions. Our longitudinal design resulted in 288 RNAseq samples across the three members, and 168 community metabolomics samples analyzed in each of four mass spectral modes (polar/nonpolar, positive/negative modes = 672 total mass spectral profiles). After quality control, we were left with 281 RNAseq and 605 total mass spectral profiles for the integrated analyses (https://github.com/ShadeLab/Paper_Chodkowski_3member_SynCom_2021/tree/master/SummaryOfSamples). First, we present a summary of experiments and cell viability (section "SynCom design/sampling scheme and membership cell viability"). Then, we present the results of general responses of transcription (section "Stationary phase transcript dynamics of microbial community members") and exometabolomics (section "Stationary phase exometabolite dynamics of microbial communities"), separately. Then, we integrate transcriptomic and metabolomic efforts to determine the upregulation of BSGCs and identify exometabolites of interest from mass spectrometry (section "B. thailandensis increases competition strategies in the presence of neighbors"). Lastly, we then present a transcriptomics co-expression network to ask how the upregulation of BSGCs influenced interspecies interactions through coordinated

longitudinal gene expression (section "Interspecies co-transcriptional networks reveal coordinated gene expression related to competition").

### SynCom design/sampling scheme and membership cell viability

We had four replicate, independent time series for each of the seven community memberships (three of each monoculture, three of each pair in coculture, and the three-member community). We define membership as the specific strains present in a given condition. Here, we focus on the multi-member analyses (two- and three-member combinations) to gain insights into community outcomes (Fig. 1A). The SynCom transwell system isolated member populations among separate transwells but permitted the exchange of their collective exometabolites *via* the plate's shared media reservoir (Fig. 1B). We collected data (transcripts, metabolites, etc) over a time series that included one exponential phase time point (12.5 h) followed by five stationary phase time points (25–45 h sampled at 5-h intervals; Fig. 1C).

We observed relatively unchanged viability in *B. thailandensis* across all conditions (Fig. S1; panels A through D). On the contrary, we observed a slight reduction (~2.1 log2 fold change) in *C. subtsugae* live cell counts, and a drastic reduction (~4.7 log2 fold change) in *P. syringae* live cell counts, when either member was cocultured with *B. thailandensis* (Fig. 2; panels A vs C and panels D vs F, respectively). Reductions in cell viability of *C. subtsugae* and *P. syringae* were also present in the 3-member community (Fig. S1; panels E and F). *C. subtsugae* and *P. syringae* had minimal effects on each other (Fig. 2; panels B and E). Dead cell accumulation of *P. syringae* plateaued in coculture conditions compared to monoculture, suggesting cell lysis (Fig. 2, panels D through F). We note that one doubling occurred in *B. thailandensis* and *P. syringae* monocultures, and in *C. subtsugae* in pairwise coculture with *P. syringae*. We elaborated on this finding as the possibility of a reductive cell division in our previous manuscript (36).

### Stationary phase transcript dynamics of microbial community members

Differentially expressed genes were determined by comparing time series transcript trajectories applying an FDR and LFC threshold (see Materials and Methods: *Quality filtering and differential gene expression analysis*). First, we compared each coculture to the monoculture control. A range of 153 to 276 genes were differentially expressed by each member in the coculture, irrespective of the identity of neighbors (Fig. S2). In addition, each member also had differential gene expression that was unique to a particular neighbor(s). Summarizing across all cocultures, 1,089/5,639 (19.3%) coding sequences (CDSs), 1,991/4,393 CDSs (45.3%), and 3,274/5,576 CDSs (58.7%). DEGs were determined for *B. thailandensis*, *C. subtsugae*, and *P. syringae*, respectively. The primary drivers of transcriptional response patterns for each member were community membership (PCoA axis 1) and time (PCoA axis 2) (Fig. 3; Table S1). Together, these data suggest that there are both general and specific consequences of neighbors for the transcriptional responses of these bacterial community members.

Temporal trajectories in member transcript profiles were generally reproducible across replicates (PROTEST analyses, Table S2). Each member had a distinct transcript profile ($0.480 \leq r2 \leq 0.778$ by Adonis; *P* value, 0.001; all pairwise false discovery rate [FDR]-adjusted *P* values, ≤0.01 except for two community memberships, Table S3). For all ordinations, community membership had the most explanatory value (Axis 1), followed by time (Axis 2), with the most variation explained by the interaction between time and membership (Table S1). Membership alone accounted for 60.6% and 77.0% of the variation explained in *C. subtsugae* and *P. syringae* analyses, respectively, and 46.3% in the *B. thailandensis* analysis (Table S1).

When included in the community, *B. thailandensis* strongly determined the transcript profiles of the other two members. For example, the inclusion of *B. thailandensis* in a coculture differentiated transcript profiles for both *C. subtsugae* and *P. syringae* (Fig. 3B and C; Fig. S3 to S5). The transcript profile differences between monoculture and coculture conditions are largest for *C. subtsugae* (Fig. S4) and *P. syringae* (Fig. S5) when

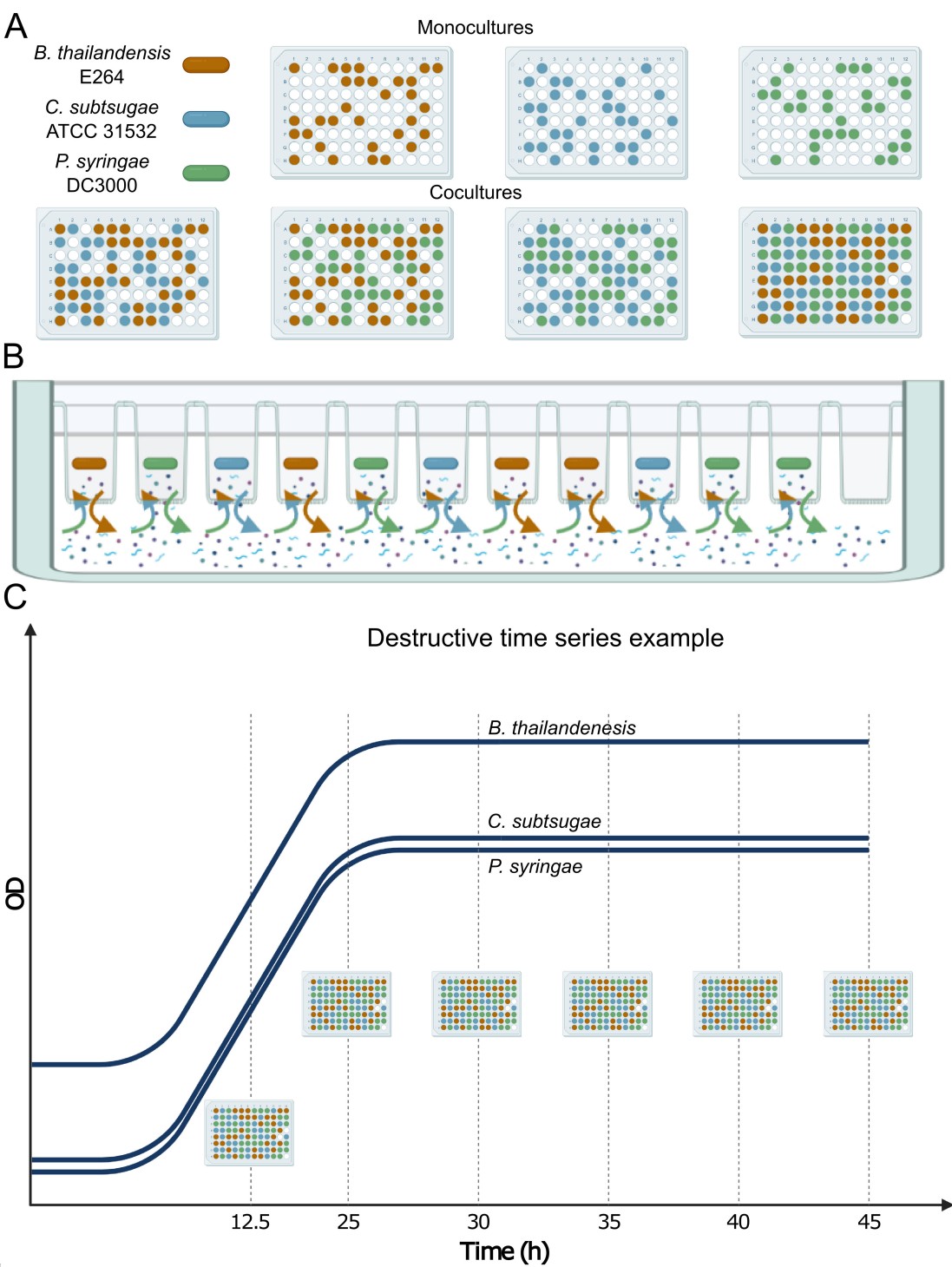

**FIG 1** Experimental design and destructive sampling procedure of transwell plates. There were seven conditions, six time points/conditions, and four independent replicates/conditions (168 total transwell plates). Each member occupied 31 wells/condition to maintain member-specific population density across all conditions (A). The SynCom transwell plate maintains the physical separation of members in individual wells while permitting exometabolite exchange through a 0.22-µm-pore filter bottom. Exometabolite exchange occurs *via* a bottom-fitted shared medium reservoir (B; (35)). Six replicate transwell plates were prepared for a time-course experiment. The time-course experiment included one exponential phase time point and five stationary phase time points. At specified time points, a transwell plate was destructively sampled (C). Note that all members were diluted to different starting ODs to allow for all members to achieve stationary phase within a two-hour window of each other. This figure was created with BioRender.com.

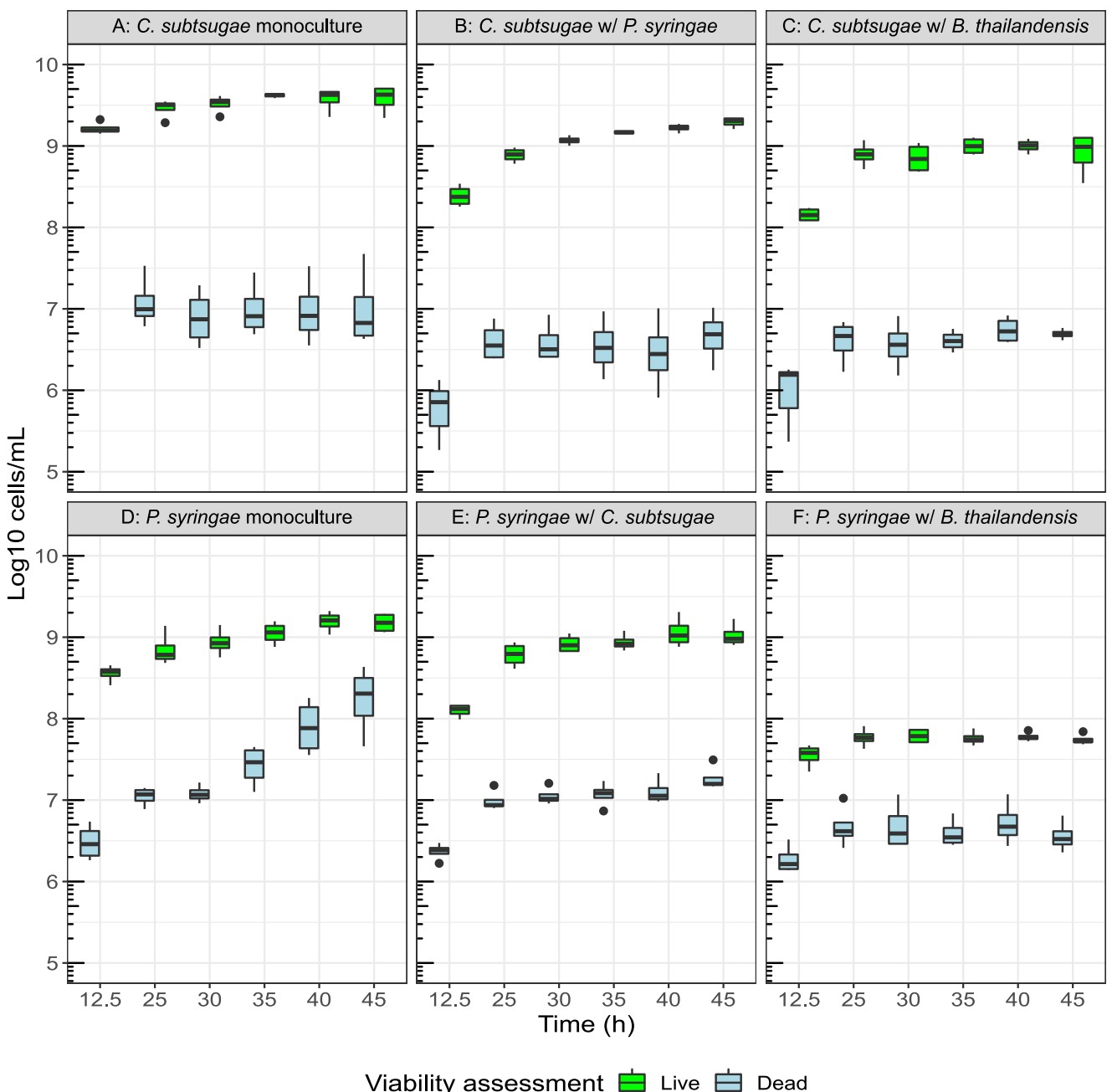

**FIG 2** Loss of cell viability in *B. thailandensis* cocultures. Live (green) and dead (blue) flow cytometry cell counts for *C. subtsugae* (Top row, panels A–C) and *P. syringae* (Bottom row, panels D–F) from Syto9- and propidium iodide-stained cells (*n* = 4 to 5 technical replicates/time point/community membership/transwell plate and *n* = 4 independent replicates/time point/community membership). Cell counts are from monocultures (panels A and D), cocultures with *P. syringae* (panel B) or *C. subtsugae* (panel E), and cocultures with *B. thailandensis* (panels C and F). The bottom and top of the box are the first (Q1) and third (Q3) quartiles, respectively, and the line inside the box is the median. The whiskers extend from their respective hinges to the largest value (top), and the smallest value (bottom) was no further away than 1.5 × the interquartile range. Points represent outliers that are less than 1.5 x the interquartile range of Q1 or greater than 1.5 x the interquartile range of Q3.

*B. thailandensis* is included in the coculture. Thus, *B. thailandensis* appears to have had a dominating influence on the transcriptional response of neighbors, and these responses were dynamic with respect to time.

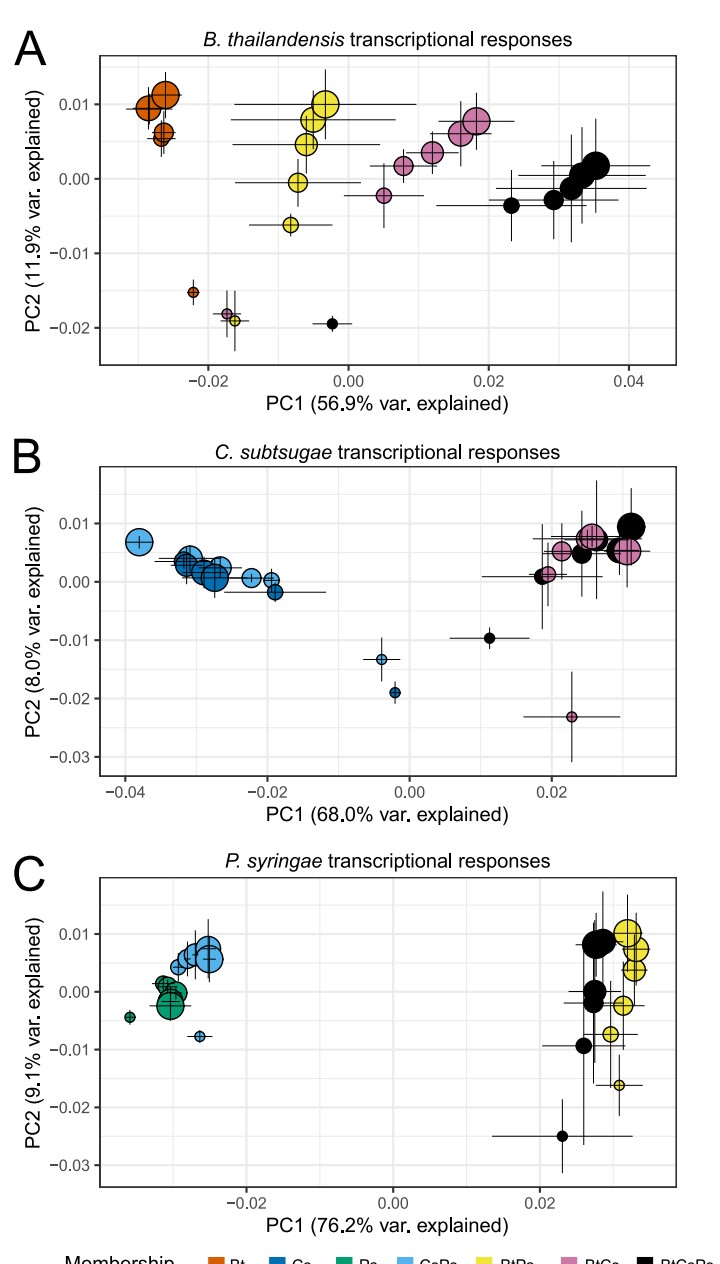

**FIG 3** Transcriptional responses are driven by community membership and time. Shown are principal coordinate analysis (PCoA) plots for *B. thailandensis* (A), *C. subtsugae* (B), and *P. syringae* (C). Each PCoA sub-panel presents the time series of transcriptional patterns of the focal member given each of its four growth conditions (one monoculture condition, two pairs, and one three-member). Each point represents a mean transcript profile for a community member given a particular condition (indicated by symbol color) and sampled at a given time point over exponential and stationary phases (in hours since inoculation, h, indicated by symbol size, *n* = 3 to 4 replicates per time point/community membership). The Bray-Curtis distance metric was used to calculate dissimilarities between transcript profiles. Error bars are one standard deviation around the mean axis scores. Note that transcriptional responses are driven by community membership on PCoA axis 1 and time on PCoA axis 2 across all plots.

We analyzed clusters of orthologous groups of proteins (COGs) to infer the responses of members to their neighbors. DEGs were categorized as upregulated or downregulated based on temporal patterns and representation in COGs (Fig. S6). We focused

on the largest differences between total DEGs upregulated and total DEGs downregulated within a COG, which provides insights into broad biological processes affected by community membership. COGs with large differences toward upregulation in *B. thailandensis* included cell motility [N], secondary metabolites biosynthesis, transport, and catabolism [Q], and signal transduction mechanisms [T] while COGs with large differences toward downregulation included defense mechanisms [V], energy production and conversion [C], translation, and ribosomal structure and biogenesis [J]. These results suggest that *B. thailandensis* responds to neighbors *via* downregulation of growth and reproduction and upregulation of secondary metabolism. We, therefore, hypothesized that *B. thailandensis* was producing bioactive exometabolites against *C. subtsugae* and *P. syringae* to competitively inhibit their growth.

Because of the strong transcript response of *C. subtsugae* and *P. syringae* when neighbored with *B. thailandensis* (Fig. 3B and C), we focused on COGs within community memberships with *B. thailandensis* (Fig. S6B and S6C, rows 2 and 3). The COG with large differences toward upregulation in both *C. subtsugae* and *P. syringae* were translation, ribosomal structure, and biogenesis [J]. COG groups tending toward downregulation in *C. subtsugae* and *P. syringae* were signal transduction mechanisms [T] and secondary metabolites biosynthesis, transport, and catabolism [Q], respectively. These results suggest that the presence of *B. thailandensis* alters its neighbor's ability to respond to the environment and inhibits secondary metabolism. The effects of *B. thailandensis* on *C. subtsugae* and *P. syringae* were also evident by mapping timeseries LFCs onto KEGG pathways. Various enzymes involved in central metabolism, fatty acid degradation, growth, transport, and response systems were upregulated when *B. thailandensis* was grown with either or both members (https://figshare.com/s/b7f5e559a32cc5c8a61f).

The above analyses focused on DEGs determined by comparing each coculture to the monoculture control. However, we also wanted to understand differences between pairs to determine whether the alterations in transcripts were attributed to specific memberships (aka interspecies interactions). A total of 436, 1,762, and 2,962 DEGs were determined when comparing the pairs including *B. thailandensis*, the pairs including *C. subtsugae*, and the pairs including *P. syringae*, respectively. We detected member-specific effects on the COGs that were differentially expressed (Fig. S7). These data suggest that there were transcriptional changes driven by particular members and given their partners. Due to the physical separation of members in our SynCom plate system, these member-specific interspecies interactions were very likely exometabolite mediated.

### Stationary phase exometabolite dynamics of microbial communities

Because member populations are physically separated in the SynCom transwell system but allowed to interact chemically, observed transcript responses in different community memberships are inferred to result from exometabolite interactions. Spent medium from the shared medium reservoir was collected from each transwell plate and analyzed using mass spectrometry to detect exometabolites. Our previous manuscript focused on exometabolite dynamics in monocultures (36). Here, we focused our analysis on those exometabolites that had maximum accumulation in a coculture (either in pairs or in a three-member community). Consistent with the transcript analysis, we found that both community membership and time explained the exometabolite dynamics and that the explanatory value of membership and time was maintained across all polarities and ionization modes (Fig. 4; Table S4).

Temporal trajectories in exometabolite profiles were generally reproducible across replicates with some exceptions (PROTEST analyses, Table S5; Supplementary File 1). Exometabolite profiles were distinct by community membership ($0.475 \leq r2 \leq 0.662$ by Adonis; *P* value, 0.001; all pairwise false discovery rate [FDR]-adjusted *P* values, ≤0.01 except for two comparisons, Table S6) and also dynamic over time. As observed for the member transcript profiles, the interaction between membership and time had the highest explanatory value for the exometabolite data (Table S4).

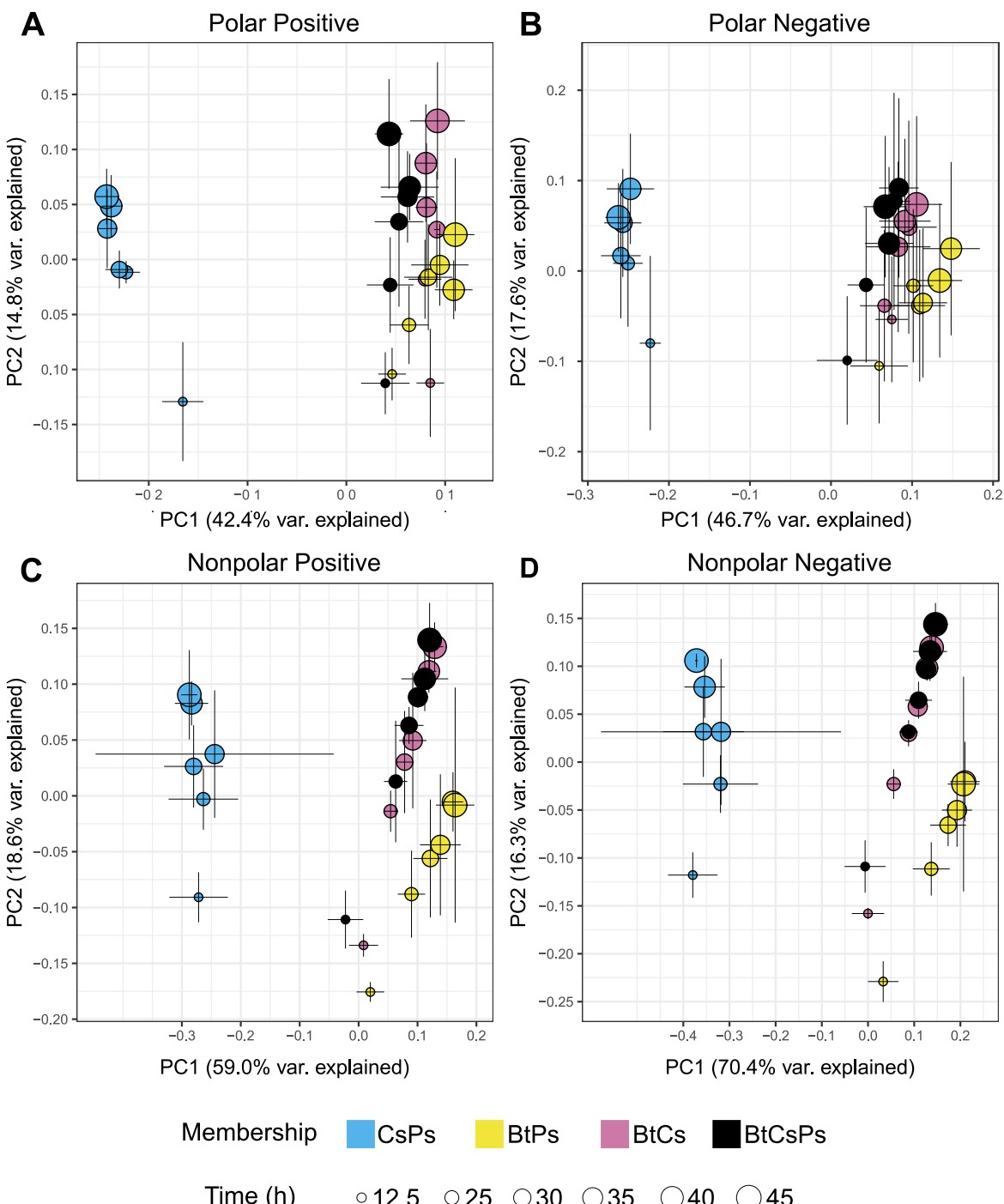

FIG 4 Bacterial community exometabolite profiles differ by community membership and time. Shown are PCoA plots for exometabolite profiles from the following mass spectrometry modes: polar positive (A), polar negative (B), nonpolar positive (C), and nonpolar negative (D). Each point represents the mean exometabolite profile (relative contributions by peak area) given a particular community membership (indicated by symbol color) at a particular time point (indicated by symbol size). The Bray-Curtis distance metric was used to calculate dissimilarities between exometabolite profiles. Error bars are 1 standard deviation around the mean axis scores ($n$ = 2 to 4 replicates). Bt is *B. thalandensis*, Cs is *C. subtsugae*, and Ps is *P. syringae*.

We found that the *C. subtsugae-P. syringae* coculture exometabolite profiles were consistently the most distinct from the other coculture memberships (Fig. 4), support-ing, again, that the inclusion of *B. thalandensis* was a major driver of exometabolite dynamics, possibly because it provided the largest or most distinctive contributions

to the community exometabolite pool. Indeed, we observed that a majority of the most abundant exometabolites were either detected uniquely in the *B. thailandensis* monoculture or accumulated substantially in its included community memberships (Fig. S8). Some exometabolites detected in *B. thailandensis*-inclusive communities were not detected in its monocultures (Fig. S8D), suggesting that the inclusion of neighbors contributed to the accumulation of these particular exometabolites (e.g., upregulation of biosynthetic gene clusters or lysis products). *C. subtsugae* and *P. syringae* contributed less to the three-member community exometabolite profile, as exometabolites detected in the *C. subtsugae*-*P. syringae* coculture were less abundant and had lower accumulation over time in the three-member community (Fig. S8A). Together, these results suggest that *B. thailandensis* can suppress or overwhelm expected outputs from neighbors.

Exometabolites categorized as primary metabolites were identified according to Metabolomics Standards Initiative (MSI) level 1 criteria (58). We identified primary metabolites accumulated in the shared medium reservoir over time in each monoculture (Fig. 5; (36)) to compare their dynamics in cocultures. These primary metabolites were detected to decrease in concentration across coculture conditions, suggesting metabolic inhibition or interspecies uptake. In addition, we also found a subset of primary metabolites that accumulate substantially in the exponential phase in monocultures (Fig. S9). Taken together, each member contributed a unique set of primary metabolites to the community exometabolite pool. The uptake and metabolism of these primary metabolites by the non-producing members may directly affect the available pool of exometabolites in cocultures, particularly with respect to exometabolites contributed from secondary metabolism.

In summary, we observed both increased accumulation and unique production of exometabolites in pairs and in the three-member community, with *B. thailandensis* contributing the most to the shared exometabolite pool as determined by comparisons with its monoculture exometabolite profile. Related, the transcriptional responses of *C. subtsugae* and *P. syringae* in the three-member community are most like their respective transcriptional response when neighbored with *B. thailandensis* alone, despite the presence of the third neighbor.

## *B. thailandensis* increases competition strategies in the presence of neighbors

Given the observed reduction in cell viability (Fig. 2) and that there have been competitive interactions between *B. thailandensis* and *C. subtsugae* previously reported (33), we hypothesized that *B thailandensis* was using competition strategies to influence its neighbors *via* production of bioactive exometabolites. If true, we would expect transcriptional upregulation in *B. thailandensis* biosynthetic gene clusters (BSGC) that encode bioactive exometabolites. Indeed, when compared to the monoculture control, we found evidence of upregulated BSGCs across various time points in stationary phase in *B. thailandensis* cocultures (Fig. 6; Table S7). Some of these upregulation patterns were associated with particular pairs of members and some upregulation patterns were strongest in the full community (e.g., thailandamide). For example, *B. thailandensis* upregulated an unidentified non-ribosomal peptide synthetase (NRPS) when paired with *P. syringae,* but when paired with *C. subtsugae*, upregulated a different BSGC encoding an unidentified beta-lactone. This suggests that *B. thailandensis* responded to neighbors by upregulating genes involved in the production of bioactive compounds, likely to gain a competitive advantage. However, not all BSGCs in *B. thailandensis* were upregulated. Some BSGCs were unaltered or downregulated (Fig. S10). *C. subtsugae* upregulated only 1 BSGC, an uncharacterized hybrid nonribosomal peptide synthetase-type I polyketide synthase, in coculture with *B. thailandensis,* while *P. syringae* did not upregulate any BSGC in any coculture (Fig. S11 and S12). Interspecies interactions led to the upregulation of BSGCs in both *B. thailandensis* and *C. subtsugae* and three of these BSGCs encode potentially novel bioactive exometabolites.

Because *B. thailandensis* upregulated the transcription of various BSGCs when grown in cocultures, we asked if this led to the unique production of or increased accumulation

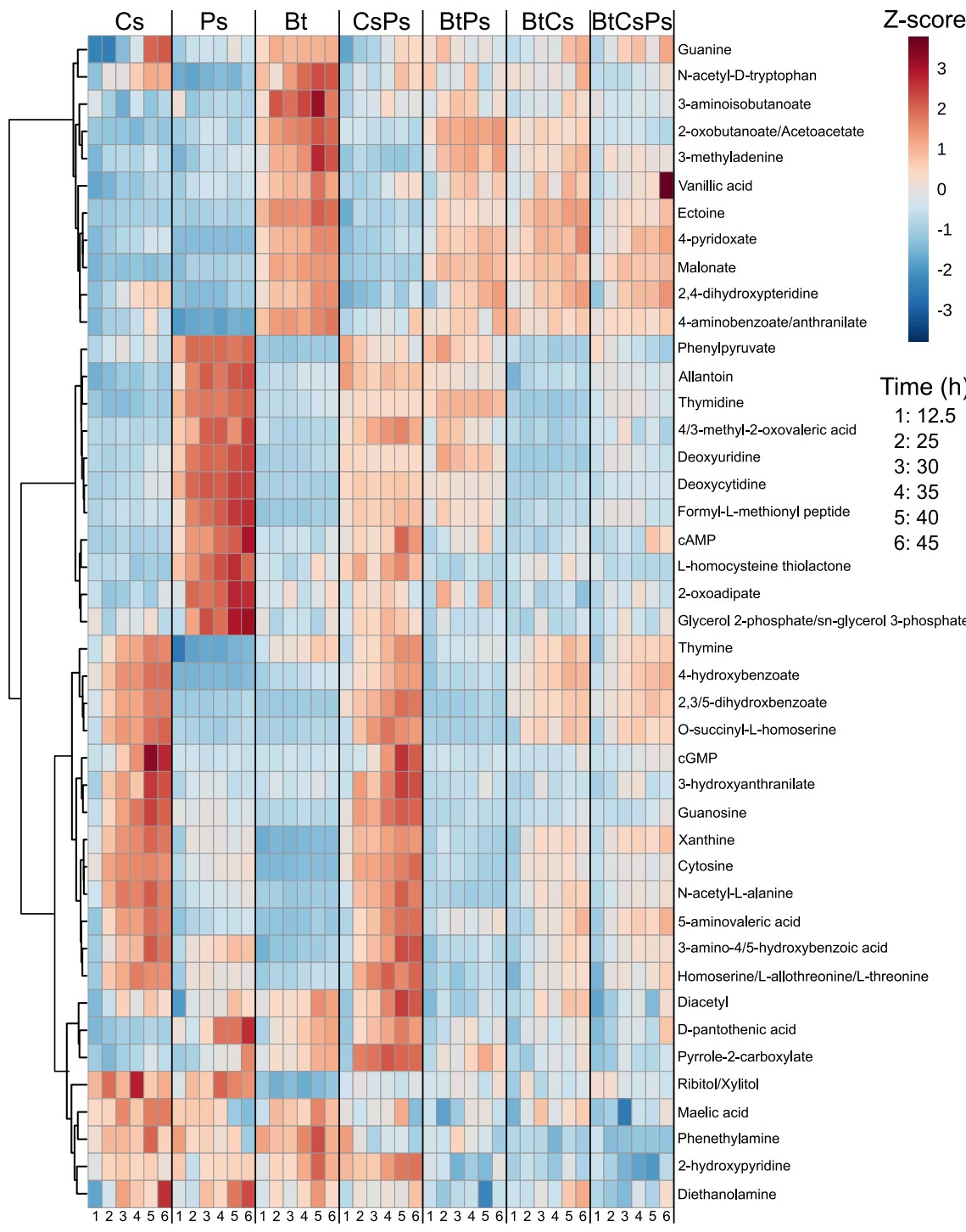

**FIG 5** Primary metabolites accumulated in monocultures have altered dynamics in cocultures. A heat map of identified, primary metabolites is shown for *C. subtsugae* monoculture (Cs), *P. syringae* monoculture (Ps), *B. thailandensis* monoculture (Bt), *C. subtsugae-P. syringae* coculture (CsPs), *B. thailandensis-P. syringae* coculture (BtPs), *B. thailandensis-C. subtsugae* coculture (BtCs), and the three-member community (BtCsPs), where samples are in columns and exometabolites are in rows. These exometabolites were filtered based on their time series accumulation in monocultures (See supplementary methods for details). Data for each sample are the averages from independent time point replicates ($n = 3$ to 4). Euclidean distance was calculated from Z-scored mass spectral profiles. Features with similar dynamics were clustered by Ward's method.

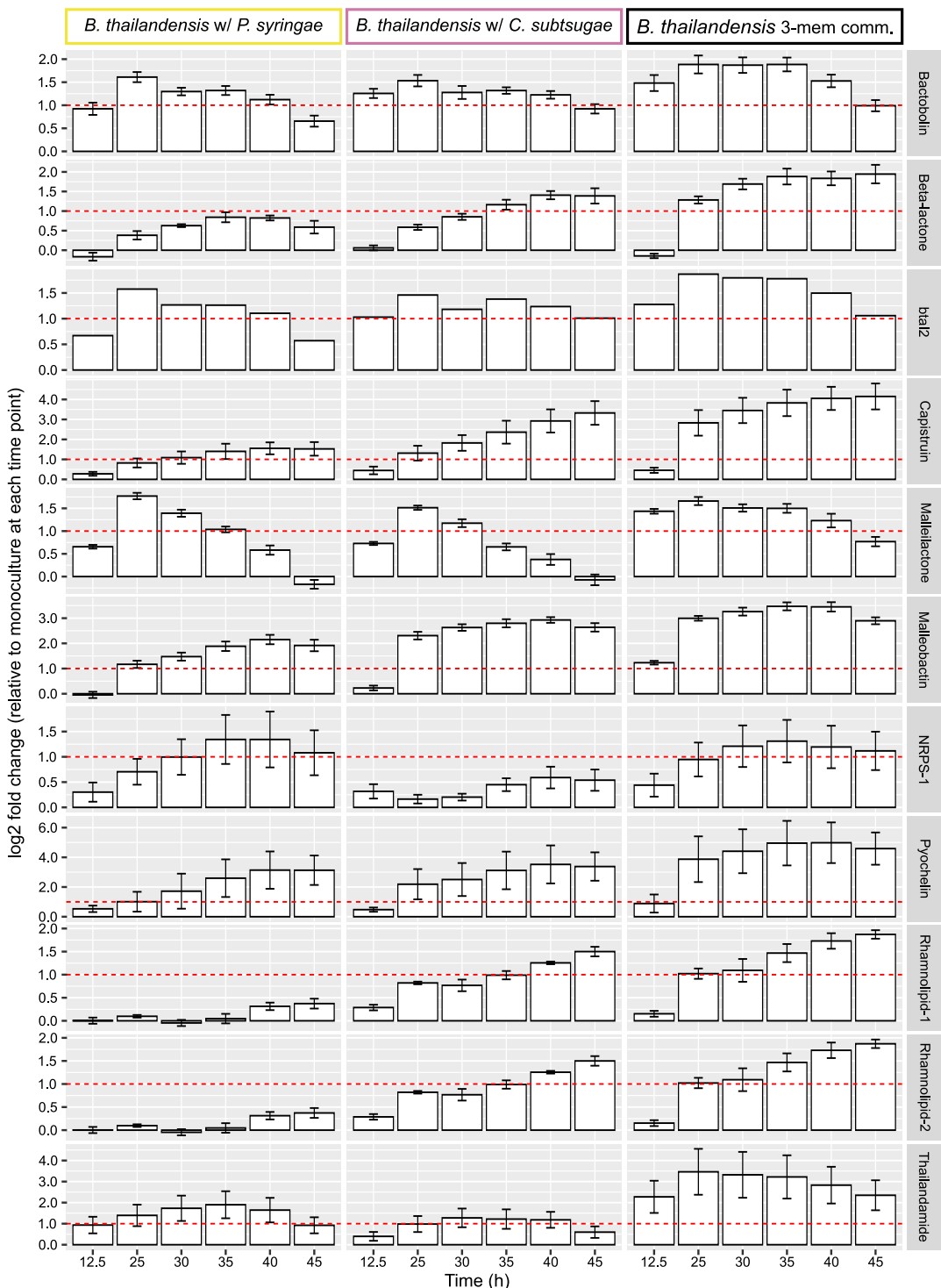

**FIG 6** *B. thailandensis* upregulates BSGC in cocultures. Columns represent community membership for *B. thailandensis* cocultures and rows represent BSGCs in *B. thailandensis* that were determined to be upregulated compared to the monoculture control. Genes part of a BSGC were curated from antiSMASH predictions and literature-based evidence. Within each BSGC at each time point, the log2 fold-change (LFC) was calculated by comparing gene counts from a coculture to the monoculture control (*n* = 3 to 4 LFC calculations/community membership/time point). Log2 fold changes were then averaged from all biosynthetic genes within the BSGC at each time point. Error bars indicate standard deviations. We defined an upregulated BSGC as a BSGC that had at least two consecutive stationary phase time points with an LFCs > 1 (indicated by the horizontal red dashed line). Note that plots for each BSGC have separate scales for the Y-axis.

of secondary metabolites as compared to when it was grown in monoculture. We identified 6 of the 11 exometabolites from the BSGCs in *B. thailandensis* that were upregulated and quantified their abundances from mass spectrometry data (Fig. 7; Supplementary File 2). We found that each identified exometabolite differentially accumulated between community memberships containing *B. thailandensis* (Table S8), particularly when comparing the *B. thailandensis* monoculture to each coculture (Table S9). As expected, these identified exometabolites were not detected in communities that did not include *B. thailandensis* (data not shown). Bactobolin was the only identified exometabolite that accumulated in monoculture to equivalent levels of accumulation in all coculture conditions. The other identified secondary metabolites were not detected or did not accumulate in monoculture, suggesting interspecies induction of secondary metabolism. Thus, in response to an exometabolite from either *C. subtsugae* or *P. syringae*, *B. thailandensis* increased its competitive strategies through the upregulation and production of many bioactive exometabolites. Of these bioactive exometabolites, three are documented antimicrobials (59–61), two are siderophores (62, 63), and one is a biosurfactant (64). We conclude that *B. thailandensis* produced bioactive exometabolites to competitively interact using both interference and exploitative competition strategies (65). Given that *B. thailandensis* upregulated competitive strategies, and responded more broadly in producing competition-supportive exometabolites when grown with neighbors, we hypothesized that these bioactive exometabolites are responsible for the altered transcriptional responses in *C. subtsugae* and *P. syringae*.

In our experimental design, we adjusted glucose concentration depending on plate occupancy. Glucose concentration increased as plate occupancy increased (31 wells vs 62 wells vs 93 wells) but a member consistently occupied 31 wells across all experimental conditions. One complication of this design is that population density and resource concentration could contribute to differences in transcripts and exometabolites in a member-agnostic manner. To address this, we performed additional SynCom experiments to affirm confidence that some changes in transcripts and exometabolites are attributable to exometabolite-mediated interspecies interactions. In these experiments, we increased the plate occupancy of *B. thailandensis* in monoculture while subsequently increasing resource concentration. Pairwise cocultures and the three-member community SynCom experiments were repeated as well (see Supplementary methods). We calculated the relative gene expression of three genes in the thailandamide operon (*thaF*, *thaK*, and *thaQ*) through RT-qPCR by comparing each experimental condition to the monoculture control (*B. thailandensis*, 31 wells in M9—0.067% glucose). Decreased gene expression was observed across all three genes as both plate occupancy and resource concentration increased in *B. thailandensis* monocultures. In fact, *thaF*, *thaK*, and *thaQ* gene expression was further reduced in the 93-well *B. thailandensis* monoculture compared to the 62-well *B. thailandensis* monoculture, suggesting that the thailandamide operon trended toward reduced expression as a function of *B. thailandensis* plate occupancy in monoculture conditions. On the contrary, *thaF*, *thaK*, and *thaQ* had increased expression in all coculture memberships, suggesting that exometabolite interspecies interactions were responsible for the increased expression of a BSGC in *B. thailandensis* (Table S10).

### Interspecies co-transcriptional networks reveal coordinated gene expression related to competition

We performed interspecies co-expression network analysis to infer interspecies interactions. We used temporal profiles from transcriptional responses to generate co-expression networks for *B. thailandensis*-*C. subtsugae* and *B. thailandensis*-*P. syringae* cocultures, respectively (Table S11). As expected, the majority of nodes in the network had intraspecies edges only, with interspecies edges comprising 1.85% and 1.90% of the total edges in the *B. thailandensis*-*C. subtsugae* and *B. thailandensis*-*P. syringae* networks, respectively. We explored interspecies edges for evidence of interspecies transcriptional co-regulation.

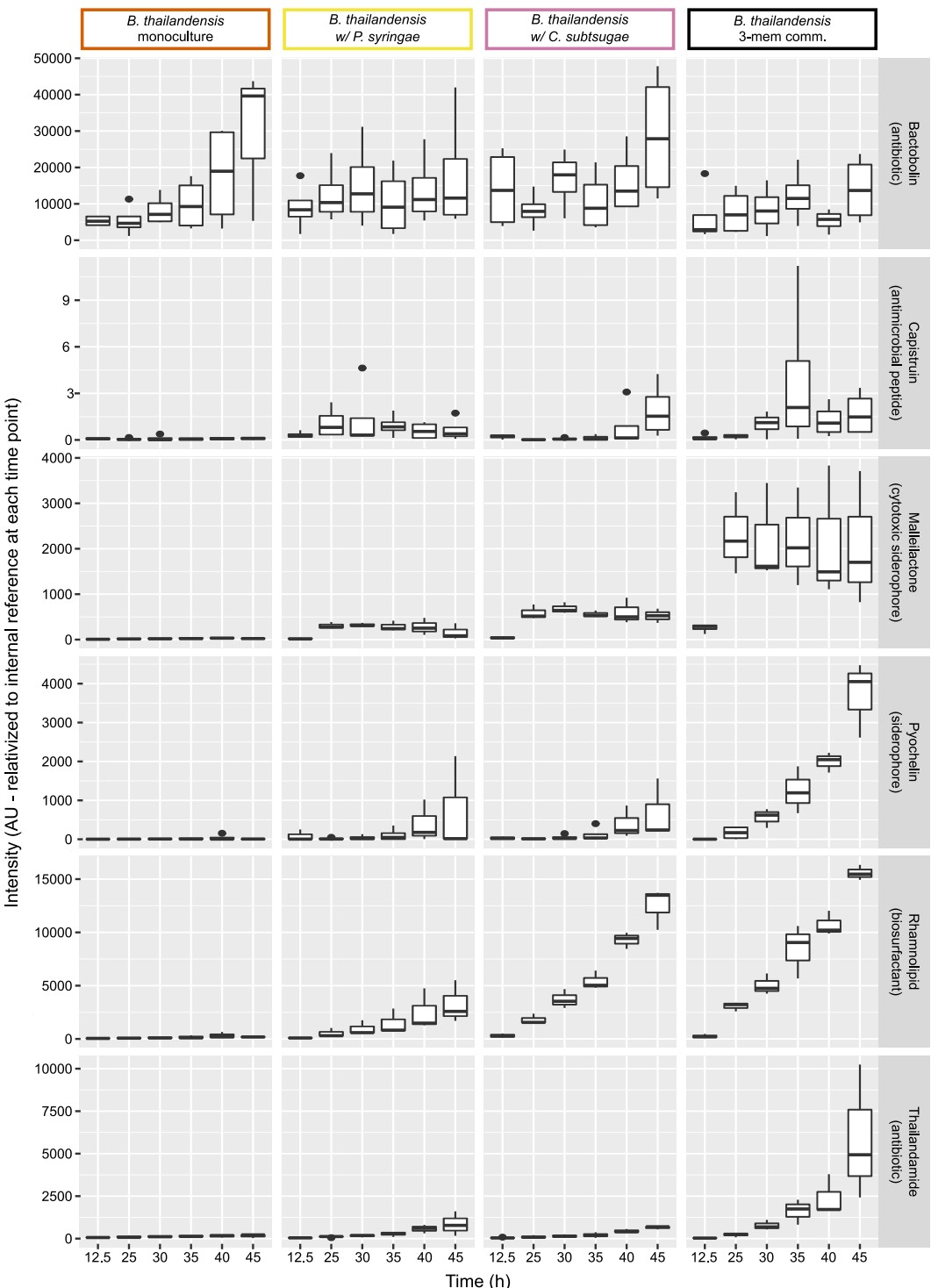

**FIG 7** Coculture upregulation of BSGCs from *B. thailandensis* translates to temporally accumulated secondary metabolites. Columns represent community membership and rows represent identified secondary metabolites in *B. thailandensis*. Known bioactive secondary metabolites produced by *B. thailandensis* were identified in MZmine 2 through the observation of MS and MS/MS data. The accumulation of each exometabolite was quantified through time ($n$ = 2 to 4 integrated peak areas per time point). The bottom and top of the box are the first (Q1) and third (Q3) quartiles, respectively, and the line inside the box is the median. The whiskers extend from their respective hinges to the largest value (top), and the smallest value (bottom) was no further away than 1.5× the interquartile range. Points represent outliers that are less than 1.5× the interquartile range of Q1 or greater than 1.5× the interquartile range of Q3.

We performed two analyses, module analysis and Gene Ontology (GO) enrichment, to validate networks and infer interspecies interactions (Fig. S13). Module analysis validated networks as intraspecies modules enriched for biological processes (Supplementary File 3). To infer interspecies interactions, we filtered genes with interspecies edges and performed enrichment analysis (Supplementary File 4). The top enriched GO term for *B. thailandensis* when paired with *C. subtsugae* was an antibiotic synthesis of thailandamide, supporting the interpretation of interference competition. Though the top enriched GO term in *B. thailandensis* when paired with *P. syringae* was bacterial-type flagellum-dependent cell motility, antibiotic synthesis of malleilactone was also enriched. Both thailandamide genes from the *B. thailandensis-C. subtsugae* network (Fig. 8) and malleilactone genes from the *B. thailandensis-P. syringae* network (Fig. S14) formed near-complete modules within their respective BSGCs. In addition, genes that were part of the BSGC modules contained interspecies edges with both *C. subtsugae* and *P. syringae*.

At least one gene from each of *B. thailandensis's* upregulated BSGCs (Fig. 6) had an interspecies edge, except for rhamnolipid. Our interpretation of this result is that, generally, *B. thailandensis's* upregulated BSGCs had co-expression patterns with genes from the other members. In the thailandamide and malleilactone modules, some of these interspecies genes were related to stress, transport, and iron-scavenging (Supplementary File 5). The top GO term for both *C. subtsugae* and *P. syringae* genes that had edges shared with *B. thailandensis* was bacterial-type flagellum-dependent motility. Other notable enriched GO processes were efflux activity for *C. subtsugae* and signal transduction for *P. syringae*. Specifically, a DNA starvation/stationary phase gene (CLV04_2968, Fig. 8), *dspA*, was within the network module that also contained thailandamide genes from the *B. thailandensis-C. subtsugae* network and a TonB-dependent siderophore receptor gene (PSPTO_1206, Fig. S14) was within the network module that also contained malleilactone genes from the *B. thailandensis-P. syringae* network. Interestingly, both CLV04_2968 and PSPTO_1206 were DEGs and downregulated when cocultured with *B. thailandensis* (Fig. S15A and S16A, respectively). In addition, the closest homolog for *dspA* in *B. thailandensis* was unaltered (BTH_I1284, Supplementary File 6) when cocultured with *C. subtsugae* (Fig. S15B) and the closest homolog to the TonB-dependent receptor in *B. thailandensis* (BTH_I2415, Supplementary File 7) was a DEG and upregulated when cocultured with *P. syringae* (Fig. S16B). Taken together, these co-expression networks revealed interspecies coordinated expression patterns. Specifically, we detected interspecies co-expression patterns related to antibiotic upregulation in *B. thailandensis*, suggesting *C. subtsugae* and *P. syringae* were sensing and responding directly to these competition strategies of *B. thailandensis*.

## DISCUSSION

Here, we used a synthetic community system to understand how exometabolomic interactions determine members' transcriptional responses and exometabolite outputs. Our experiment used a systems approach to compare the seven possible community memberships of three members, and their dynamics in member transcripts and community exometabolites over stationary phase. Differential gene expression across community memberships and over time show that the exometabolites released by a member were sensed and responded to by their neighbors. Furthermore, members' outputs in monocultures changed because of coculturing, as evidenced by differential exometabolite production. The largest transcriptional alterations in *C. subtsugae* and *P. syringae* occurred when cocultured with *B. thailandensis*. Global expression patterns in *C. subtsugae* and *P. syringae* when in the three-member community still resembled expression patterns in pairwise cocultures with *B. thailandensis*. These transcriptional alterations in *C. subtsugae* and *P. syringae* were coordinated with increases in *B. thailandensis* competitive strategies (evaluated by BSGC transcript upregulation and exometabolite abundance). That interactions within a relatively simple community altered the transcriptional responses and exometabolite outputs of each member are important because these kinds of alterations could, in turn, drive changes in community structure

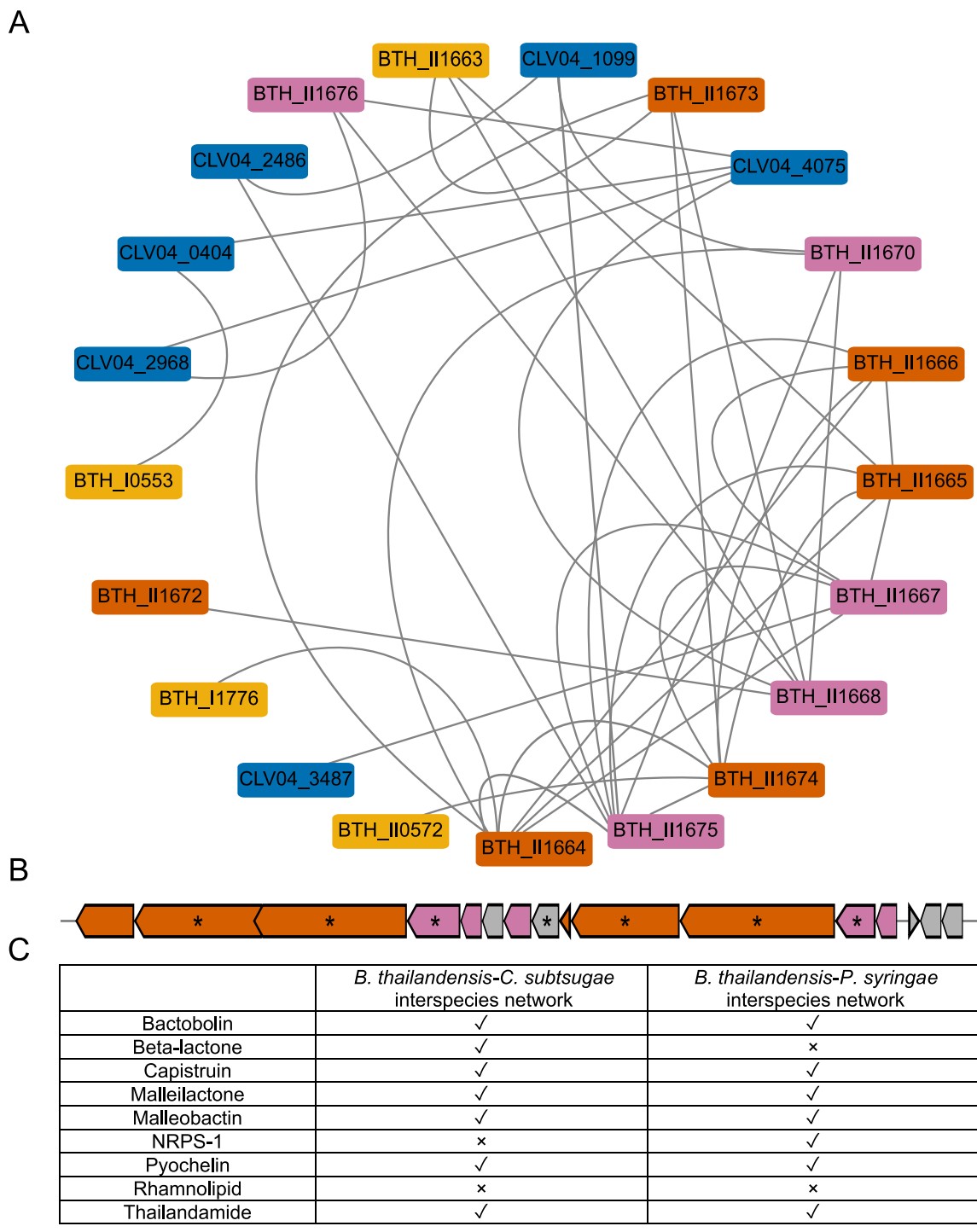

**FIG 8** *B. thailandensis* genes involved in thailandamide production are co-expressed with *C. subtsugae* genes. A network module containing the thailandamide BSGC is shown (A). The network module nodes are color-coded by according to the following criteria: thailandamide biosynthetic genes that had interspecies edges (magenta), thailandamide biosynthetic genes that did not have interspecies edges (orange), other *B. thailandensis* genes that were not part of the BSGC (yellow), and genes that were from *C. subtsugae* (blue). The chromosomal organization of the thailandamide BSGC is shown below the network module (B). The same colors are applied to the BSGC operon. The operon also depicts genes that were not detected within the interspecies network, shown in gray. Asterisks indicate core biosynthetic genes in the BSGCs, as predicted from antiSMASH. Table (C) shows upregulated *B. thailandensis* BSGCs (Fig. 6) and whether interspecies edges were detected (check is yes, x is no).

and/or function in an environmental setting. For example, it was shown that interspecies interactions more strongly influenced the assembly of *C. elegans* gut communities than host-associated factors (66). Therefore, mechanistic and ecological characterization of interspecies interactions will inform as to the principles that govern emergent properties of microbial communities.

Overall, competitive interactions predominated in this synthetic community. This was first evidenced by reductions in viable cell counts in both *C. subtsugae* and *P. syringae* when cocultured with *B. thailandensis*. Interestingly, *P. syringae* was the only member to have an exponential increase in dead cell counts in monoculture. *P. syringae* dead cell count accumulation ceased in coculture conditions. We attribute this finding to the overall reduction of cell viability and/or lysis of dead cells when cocultured.

Our previous study found that, over the stationary phase in monocultures, each member released and accumulated at least one exometabolite documented to be involved in either interference or exploitative competition (36). This suggests that entry into stationary phase primed members for competitive interactions, regardless of heterospecific neighbors present. We interpret this strategy of preemptive aggression to be especially advantageous to *B. thailandensis,* as it successfully used competitive strategies against both *C. subtsugae* and *P. syringae. B. thailandensis's* success was supported by decreased viable *P. syringae* cells when cocultured with *B. thailandensis*. Though *C. subtsugae* viable cell counts were not as affected directly by the coculture with *B. thailandensis*, *B. thailandensis*-produced bactobolin (67) was detected in the shared medium reservoir. Bactobolin is a bacteriostatic antibiotic previously shown to be bioactive against *C. subtsugae* (33) through ribosome binding (59). However, *C. subtsugae* can resist bactobolin through upregulation of an RND-type efflux pump (68). This finding also is supported by our data, as all genes coding for the CdeAB-OprM RND-type efflux system were DEGs and upregulated in *C. subtsugae* cocultures with *B. thailandensis* (CLV04_2413-CLV04_2415).

When cocultured with *B. thailandensis*, we observed COG groups such as translation, ribosomal structure, and biogenesis [J] had large differences toward upregulation in both *C. subtsugae* and *P. syringae*. At first glance, this seems at odds with our interpretation of *B. thailandensis* competitiveness toward *C. subtsugae* and *P. syringae*. In other words, how is *B. thailandensis* effectively competing *via* interference competition whether both *C. subtsugae* and *P. syringae* are upregulating machinery for growth? There are both theoretical (69) and experimental (70) evidence that show how cells treated with antibiotics stimulate ribosomal production to maintain a sufficient number of active ribosomes. As previously mentioned, *B. thailandensis*-produced bactobolin binds to the ribosome and can inhibit *C. subtsugae* (33, 59). We also have evidence that bactobolin inhibits *P. syringae* (data not shown). It could be that bactobolin is stimulating ribosomal production in *C. subtsugae* and *P. syringae* as a survival mechanism to maintain protein production by maintaining enough active ribosomes. There was also evidence of *B. thailandensis* antibiotic efficacy against *C. subtsugae* and *P. syringae*, including general loss of cell viability and upregulation of various enzymes involved in central metabolism by both members when they were cocultured with *B. thailandensis* (https://figshare.com/s/b7f5e559a32cc5c8a61f). These patterns are consistent with antibiotic treatments in *Escherichia coli* and *Staphylococcus aureus* where the upregulation of oxidative phosphorylation due to drug treatment contributes to antibiotic efficacy (71, 72). A barrage of *B. thailandensis*-produced antibiotics (Fig. 6 and 7) likely drove the transcriptional patterns in *C. subtsugae* and *P. syringae*.

Coculturing can induce secondary metabolism (73–75) because an exometabolite produced by one microbe can prompt secondary metabolism in a neighbor (31). We found that coculturing led to the upregulation of numerous BSGCs in *B. thailandensis*. These exometabolites included bactobolin, malleilactone (62, 76; siderophore and cytotoxin), malleobactin (77, 78; siderophore), capistruin (79; lasso peptide), thailandamide (80; polyketide), pyochelin (63; siderophore), rhamnolipids (64; biosurfactants), and two uncharacterized BSGCs encoding nonribosomal peptide synthetases. Of these

exometabolites, bactobolin, capistruin, and thailandamide have documented antimicrobial activities through translation inhibition (59), transcription inhibition (60), and inhibition of fatty acid synthesis (61), respectively. For those exometabolites we were able to identify with mass spectrometry, their accumulation in cocultures was correlated with the upregulation of their BSGCs. Furthermore, up/downregulated patterns across all *B. thailandensis* BSGCs were consistent with ScmR global regulatory patterns of secondary metabolism (81).

We acknowledge that this study is limited in its ability to pinpoint the underlying mechanisms driving the activation of secondary metabolism, particularly in *B. thailandensis*. Aside from self-activating mechanisms documented in *B. thailandensis* (e.g., quorum-sensing-driven bactobolin production) and/or sensing antibiotics and competitively responding (82), we note two major patterns in exometabolite production in the monocultures that may have contributed to the activation of secondary metabolism in the cocultures. First, each member released and accumulated a unique set of primary metabolites over their time series. These exometabolites had relatively reduced concentrations in their coculture conditions. Second, because our experimental design included a comparative time point taken during exponential growth, we also identified a unique set of primary metabolites that had substantially accumulated by 12.5 h. Indeed, primary metabolites (83) have been documented to induce secondary metabolism in *B. thailandensis*. Thus, it is possible that the dynamics observed over the stationary phase could be attributed also to the uptake of exometabolites that were produced earlier in the exponential phase, or to the uptake of accumulated primary metabolites. Instead of pinpointing single molecule elicitors of secondary metabolism, our systems-level approach is better used to improve understanding of the environmental and ecological factors that contribute to member or community success.

*C. subtsugae* can inhibit *B. thailandensis* (33) but we did not observe *B. thailandensis* inhibition based on cell counts. However, we did find that in stationary phase *C. subtsugae*-*B. thailandensis* cocultures, *C. subtsugae* upregulated an uncharacterized hybrid nonribosomal peptide synthetase-type I polyketide synthase. *P. syringae* was the least competitive of the three neighbors, as evidenced by a reduction in live cell counts when cocultured with *B. thailandensis*. Also, *P. syringae* did not increase competitive strategies when cocultured, as no BSGCs were upregulated across all coculture conditions. In summary, though all three neighbors had a potential to use competitive strategies and maintained competitive strategies in monoculture (36), *B. thailandensis* was most successful in cocultures over the stationary phase through increased production of exometabolites involved in interference and exploitative competition strategies.

Given the upregulation of BSGCs in *B. thailandensis* and the strong transcriptional responses of *C. subtsugae* and *P. syringae* to the presence of *B. thailandensis*, we hypothesized that competitive exometabolites were contributing to their community dynamics. Thus, we used a co-expression network analysis with our longitudinal transcriptome series to infer interspecies interactions (84). The use of this approach was first demonstrated to infer coregulation between a phototroph-heterotroph commensal pair (85). Our network confirmed that *B. thailandensis* BSGCs had coordinated gene expression patterns with both *C. subtsugae* and *P. syringae*. Interspecies nodes in both networks contained various genes involved in the upregulated *B. thailandensis* BSGCs. We focused on interspecies edges within thailandamide nodes for the *B. thailandensis*-*C. subtsugae* network and interspecies edges within malleilactone nodes for the *B. thailandensis*-*P. syringae* network because these were significantly enriched as interspecies nodes. A *C. subtsugae* gene of interest, CLV04_2968, was contained within the thailandamide cluster of interspecies nodes. This gene codes for a DNA starvation/stationary phase protection protein and had the highest homology to the Dps protein in *Escherichia coli* across all *C. subtsugae* protein-coding genes. Dps mediates tolerance to multiple stressors and *dps* knockouts are more susceptible to thermal, oxidative, antibiotic, iron toxicity, osmotic, and starvation stressors (86). Interestingly, CLV04_2968 was downregulated when cocultured with *B. thailandensis*, suggesting that

*B. thailandensis* attenuates *C. subtsugae* stress tolerance over the stationary phase. While we observed a slight decrease in viable *C. subtsugae* cells when cocultured with *B. thailandensis*, one may expect *C. subtsugae* to have increased sensitivity to subsequent stress (e.g., pH stress; 87) resulting from CLV04_2968 downregulation in the presence of *B. thailandensis*.

In the *B. thailandensis-P. syringae* co-expression network, a *P. syringae* gene of interest, PSPTO_1206, was contained within the malleilactone cluster of interspecies nodes. PSPTO_1206 is annotated as a TonB-dependent siderophore receptor. A *P. syringae* iron-acquisition receptor had coordinated expression with malleilactone, which has been characterized as a siderophore with antimicrobial properties (62). Interestingly, this gene was downregulated when in coculture with *B. thailandensis*. By contrast, the closest TonB-dependent siderophore receptor homolog to PSPTO_1206 in *B. thailandensis*, BTH_I2415, was upregulated in coculture conditions with *P. syringae*. To summarize, co-expression network analysis revealed interspecies coordinated gene expression patterns. Though determining directionality was beyond the scope of this analysis, we observed *B. thailandensis*-increased competition strategies were coordinated with a potential decrease in competition strategies in *C. subtsugae via* reduced stress tolerance and in *P. syringae* with reduced iron acquisition ability.

One feature of our study is that we adjusted glucose concentration depending on plate occupancy. Glucose concentration increased as membership increased but a member consistently occupied 31 wells across all experimental conditions. One could argue that resource concentration contributed to differences in transcripts and exometabolites and not interspecies interactions. However, DEGs were present when comparing pairwise coculture conditions and these were attributed to differences in temporal regulation of COG categories (Fig. S7). More specifically, regarding BSGCs, an unidentified NRPS was upregulated in *B. thailandensis* when cocultured with *P. syringae* but not when cocultured with *C. subtsugae* (Fig. 6) and, an unidentified NRPS-Type I polyketide synthase was upregulated in *C. subtsugae* when cocultured with *B. thailandensis* but not when cocultured with *P. syringae* (Fig. S11). These differences occurred in experimental conditions where the glucose concentration was the same. Furthermore, we performed additional SynCom experiments where we increased the plate occupancy of *B. thailandensis* in monoculture while subsequently increasing resource concentration. Decreased gene expression was observed across all three RT-qPCR tested thailandamide genes as both plate occupancy and resource concentration increased in *B. thailandensis* monocultures. These same three genes had increased gene expression across all cocultures. These findings show that some undefined exometabolite interspecies interactions were responsible for the increased expression of a BSGC in *B. thailandensis*. Overall, we acknowledge that resource concentration and exometabolite output are intertwined, and subsequent work could test how initial resource availability determines SynCom outcomes.

A major goal in microbial ecology is to predict community dynamics for purposes of modulating and/or maintaining ecosystem function (88, 89). At its core, microbial functional properties emerge, in part, from the concerted interactions of multi-species assemblages. The SynCom system provides a tractable experimental system to understand the relationships between exometabolite interactions and environmental stimuli to inform higher-order community interactions. Higher-order interactions are those that are unexpected based on interactions observed in simpler situations (e.g., of member pairs) (90–92). Therefore, integrating different system variables, like transcriptome and metabolome dynamics, within controlled microbial communities will inform how unexpected phenomena arise and how they contribute to deviations in predictive models of community outcomes.

Our results indicated that competition strategies were maintained despite stagnant population growth. *B. thailandensis* upregulated various bioactive exometabolites involved in both interference and exploitative competition when with neighbors. An effective competitor is often defined as its ability to outcompete neighbors *via* growth

advantage that stems from efficient nutrient uptake and/or biomass conversion rates (93, 94). We add to this that a competitor can also have a fitness advantage through effective maintenance, which can similarly employ interference or exploitative competitive strategies despite no net growth. Maintenance may ensure survival in some environments that impose a stationary phase lifestyle, where long periods of nutrient depletion are punctuated with short periods of nutrient flux. In these scenarios, it warrants to understand how competitive strategies are deployed in the interim of growth and the extent to which these interactions contribute to long-term community outcomes. Though population levels remain constant, sub-populations of growing cells have been observed in the stationary phase (95), and continued production of competitive exometabolites may serve as an advantageous strategy to hinder the growth of competitors. In addition, some antibiotics remain effective in non-replicating bacteria (96). The ability for continued maintenance *via* effective competition strategies during the stationary phase may provide spatiotemporal maintenance of population levels before growth resumption (97). Alternatively, both growth and non-growth strategies may be occurring simultaneously (e.g., as can occur in biofilms). The heterogeneity of biofilms may provide an environment where a bacterial population contains both stationary cells in the center of the colony with growing cells at the periphery of the colony that compete and alter the developmental patterns of neighboring populations (98, 99). Thus, we expect that insights into the long-term consequences of competition for microbial community outcomes will be gained by considering competition in both active growth and maintenance scenarios.

## ACKNOWLEDGMENTS

We thank Katherine B. Louie and Benjamin P. Bowen for the support in mass spectral analysis.

This material is based upon work supported by the National Science Foundation under grant DEB 1749544 and by Michigan State University. In addition, metabolite analysis and transcript sequencing were provided by a DOE-JGI Community Science Program award (proposal identifier 502921). The work conducted by the U.S. Department of Energy Joint Genome Institute, a DOE Office of Science User Facility, is supported under contract number DE-AC02-05CH11231. J.L.C. was supported by the Eleanor L. Gilmore Fellowship from the Department of Microbiology and Molecular Genetics.

J.L.C. and A.S. conceived of and designed the study. J.L.C. performed the research and analyses. J.L.C. and A.S. wrote the manuscript.

## AUTHOR AFFILIATIONS

[1]Department of Microbiology and Molecular Genetics, Michigan State University, East Lansing, Michigan, USA
[2]Universite Claude Bernard Lyon 1, Laboratoire d'Ecologie Microbienne, UMR CNRS 5557, UMR INRAE 1418, VetAgro Sup, Villeurbanne, France

## AUTHOR ORCIDs

John L. Chodkowski ⓘ http://orcid.org/0000-0001-7785-8660
Ashley Shade ⓘ http://orcid.org/0000-0002-7189-3067

## FUNDING

| Funder | Grant(s) | Author(s) |
|---|---|---|
| National Science Foundation (NSF) | 1749544 | Ashley Shade |

## AUTHOR CONTRIBUTIONS

John L. Chodkowski, Conceptualization, Data curation, Formal analysis, Investigation, Methodology, Visualization, Writing – original draft, Writing – review and editing | Ashley Shade, Conceptualization, Funding acquisition, Investigation, Project administration, Supervision, Visualization, Writing – original draft, Writing – review and editing

## DATA AVAILABILITY

Genomes for B. thailandensis, C. subtsugae, and P. syringae are available at the National Center for Biotechnology Information (NCBI) under accession numbers NC_007651 (Chromosome I)/NC_007650 (Chromosome II), NZ_CP142381, and NC_004578 (Chromosome)/NC_004633 (Plasmid A)/NC_004632 (Plasmid B), respectively. An improved annotated draft genome of C. subtsugae is available under NCBI Bio-Project accession number PRJNA402426 (GenBank accession number PKBZ00000000). Data for resequencing efforts for B. thailandensis and P. syringae are under NCBI BioProject accession numbers PRJNA402425 and PRJNA402424, respectively. Metabolomics data and transcriptomics data are also available at the JGI ta are also available at the JGI Genome Portal (100) under JGI proposal identifier 502921. MZmine XML parameter files for all analyses can be viewed at and downloaded from GitHub (see Dataset 7 at https://github.com/ShadeLab/Paper_Chodkowski_MonocultureExometabolites_2020/tree/master/Datasets). Large data files (e.g., MZmine project files) are available upon request. Supplementary files are also available on GitHub (https://github.com/ShadeLab/Paper_Chodkowski_3member_SynCom_2021/tree/master/Supplementary_Files). Computing code, workflows, and data sets are available at (https://github.com/ShadeLab/Paper_Chodkowski_3member_SynCom_2021). R packages used during computing analyses included DEseq2 (41), ImpulseDE2 (42), VennDiagram (43), ggplot2 (44), vegan 2.5-4 (45), RVAideMemoire (46), Minet (50), rtracklayer (101), viridis (102), and helper functions (103–106).

## ADDITIONAL FILES

The following material is available online.

### Supplemental Material

**Data File S1 (mSystems00064-24-s0001.xlsx).** Pairwise PROTEST analyses comparing the reproducibility of exometabolome profiles across biological replicate time series.
**Data File S2 (mSystems00064-24-s0002.xlsx).** Identification of *B. thailandensis* bioactive exometabolites of interest through observation of mass spectrometry data.
**Data File S3 (mSystems00064-24-s0003.xlsx).** Genes part of the interspecies network, their corresponding modules, and GO enrichment analysis of modules.
**Data File S4 (mSystems00064-24-s0004.xlsx).** GO enrichment analysis on genes with interspecies edges from network analysis.
**Data File S5 (mSystems00064-24-s0005.xlsx).** Gene annotations for *C. subtsugae* and *P. syringae* that contained interspecies edges with *B. thailandensis* thailandamide and malleilactone biosynthetic genes, respectively.
**Data File S6 (mSystems00064-24-s0006.txt).** Protein alignment of the DNA starvation/stationary phase protein from *C. subtsugae* and the closest homolog in *B. thailandensis*.
**Data File S7 (mSystems00064-24-s0007.txt).** Protein alignment of the TonB-dependent siderophore receptor family protein from *P. syringae* and the closest homolog in *B. thailandensis*.
**Supplemental Information (mSystems00064-24-s0008.docx).** Supplemental materials and methods, figures, and tables.

## Open Peer Review

**PEER REVIEW HISTORY (review-history.pdf).** An accounting of the reviewer comments and feedback.

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
