## [Reviewer comments · mSystems]

Bioactive exometabolites drive maintenance competition in simple bacterial communities

John Chodkowski and Ashley Shade

Corresponding Author(s): Ashley Shade, CNRS Delegation Alpes

Review Timeline:

Submission Date:

January 18, 2024

Accepted:

February 19, 2024

Editor: Kiran Patil

Reviewer(s): The reviewers have opted to remain anonymous.

Transaction Report:

DOI: <https://doi.org/10.1128/mSystems.00064-24>

Re: mSystems00064-24 (Bioactive exometabolites drive maintenance competition in simple bacterial communities)

Dear Prof. Ashley Shade:

Your manuscript has been accepted, and I am forwarding it to the ASM production staff for publication. Your paper will first be checked to make sure all elements meet the technical requirements. ASM staff will contact you if anything needs to be revised before copyediting and production can begin. Otherwise, you will be notified when your proofs are ready to be viewed.

Cover Image Submissions: If you would like to submit a potential Featured Image, please email a file and a short legend to msystems@asmusa.org. Please note that we can only consider images that (i) the authors created or own and (ii) have not been previously published. By submitting, you agree that the image can be used under the same terms as the published article. Image File requirements: TIF/EPS, 7.5 inches wide by 8.25 inches tall (at least 2,250 pixels wide by 2,475 pixels tall), minimum 300 dpi resolution (600 dpi preferred), RGB, and no figure elements, e.g., arrows or panel labels. The legend should be a short description of the image, 1-2 sentences recommended.

Sincerely,
Kiran Patil

Reviewer #2 (Comments for the Author):

We thank the authors for having revised their manuscript. In particular, the summary figure 1 was much appreciated, as it eased our understanding of the manuscript even at the second reading.

Generally, we appreciated the many steps made towards more readability in this new version of the manuscript. Concerning the main figures, the authors have decided to avoid processing the data too much to stay faithful to it, and we understand this choice. Thus, the paper remains a somewhat technical read, inherently due to the complex system of study. To our sense the main message, filling a gap in knowledge, will still be interpretable to a larger audience: that interaction between bacteria continues during stationary phase, with some species toxifying the environment for others. The evidence for this is convincing to us.

Aside from readability, we had two main comments for the manuscript. In the first one (R2.1) we suggested testing the specific metabolites in monocultures. The authors explain that only few metabolites are available, and for those available references exist that are cited in the manuscript. In addition to this, there is a possibility of crosstalk between those metabolites, hindering the general interpretation. This is generally now mentioned in the discussion part, and we believe that specialists will know the effects of those molecules in the references cited by the authors. We are therefore satisfied with this modification.

Our second concern was regarding the role of exponential phase metabolites and their potential effect in stationary phase: this concern has been fully addressed in the main text and discussion, made clear with figure 1, so is no longer a problem.